# p63: A Master Regulator at the Crossroads Between Development, Senescence, Aging, and Cancer

**DOI:** 10.3390/cells14010043

**Published:** 2025-01-03

**Authors:** Lakshana Sruthi Sadu Murari, Sam Kunkel, Anala Shetty, Addison Bents, Aayush Bhandary, Juan Carlos Rivera-Mulia

**Affiliations:** 1Department of Biochemistry, Molecular Biology and Biophysics, University of Minnesota Medical School, Minneapolis, MN 55455, USA; sadum001@umn.edu (L.S.S.M.); kunke145@umn.edu (S.K.); addisonbents01@gmail.com (A.B.); bhandaryaayush@gmail.com (A.B.); 2Stem Cell Institute, University of Minnesota Medical School, Minneapolis, MN 55455, USA; 3Masonic Cancer Center, University of Minnesota Medical School, Minneapolis, MN 55455, USA; 4Institute on the Biology of Aging and Metabolism, University of Minnesota Medical School, Minneapolis, MN 55455, USA

**Keywords:** *TP63*, senescence, aging, disease, genome organization

## Abstract

The p63 protein is a master regulatory transcription factor that plays crucial roles in cell differentiation, adult tissue homeostasis, and chromatin remodeling, and its dysregulation is associated with genetic disorders, physiological and premature aging, and cancer. The effects of p63 are carried out by two main isoforms that regulate cell proliferation and senescence. p63 also controls the epigenome by regulating interactions with histone modulators, such as the histone acetyltransferase p300, deacetylase HDAC1/2, and DNA methyltransferases. miRNA-p63 interactions are also critical regulators in the context of cancer metastasis. This review aims to elaborate on the diverse roles of p63, focusing on disease, development, and the mechanisms controlling genome organization and function.

## 1. Introduction

The tumor protein 63 gene TP63 encodes for the protein p63. Along with other tumor suppressors, p53 and p73, p63 is a member of the family of transcription factors that play critical roles in embryonic development, tissue maintenance, cancer, aging, and genome organization. This master regulator functions through distinct mechanisms that include direct interactions with specific sites in the genome and indirectly via interactions with other proteins, chromatin remodelers, or miRNA molecules [1,2]. Dissection of regulatory mechanisms has been challenging since this gene generates multiple isoforms driven by two alternative promoters and complex alternative splicing events [3,4,5]. Recent advances have unveiled a key role of p63 in development and cancer progression, which are mediated by the capacity of this transcription factor to control chromatin accessibility and enhancer activity and cellular transdifferentiation [1,6,7,8]. p63 also controls apoptosis and cell cycle during development, as well as in adult tissues. In fact, both p63 and p53 act as pro-apoptotic factors to regulate the programmed cell death of sympathetic neurons to compensate for their overproduction during development, and the full-length TAp63 isoform mediates this process [6,9]. However, regulation through p63 in development is more complex than programmed apoptosis by p53 in specific tissues and plays a vital role in lineage specification for epithelial tissues. In fact, given the diverse roles of p63 during development, distinct mutations of the *TP63* gene have been causally linked to multiple human diseases [10,11]. Point mutations in the *TP63* gene result in distinct syndromes: ectodermal dysplasia (ED) syndromes, which are ectrodactyly ectodermal dysplasia cleft lip/palate (EEC), ankyloblepharon–ectodermal dysplasia clefting (AEC), acro–dermato–ungual–lacrimal–tooth (ADULT), and limb–mammary syndrome (LMS). Developmental control mechanisms are still poorly understood, but evidence is being unveiled in mouse models. Furthermore, p63 is also required for both male and female germline maintenance [12,13,14,15,16,17]. In adult tissues, p63 is also pivotal for maintaining epidermal homeostasis [18,19]. As a transcription factor, it regulates cellular pathways required for adult tissue survival, self-renewal, and differentiation [1,2].

This review discusses the complex roles of p63 in the development and maintenance of adult tissues, alterations in cancer, aging, and aging-related disorders, and its role in regulating genome organization and function. We emphasize its multifaceted role in regulating multiple physiological processes, such as maintaining stemness in organ systems, orchestrating the cell cycle, and chromatin remodeling via crosstalk with chromatin and epigenetic modulators.

## 2. *TP63* Structure

The *TP63* spans multiple exons, and its expression is driven by alternative promoters that generate two major p63 isoforms: TAp63 and the N-terminally truncated ΔNp63 (Figure 1) [3,4,5,20]. Moreover, additional variants are generated by complex alternative splicing: α, β, γ, δ, and ε [5,20,21]. The general protein structure of p63, with some variation depending on isoform, consists of an N-terminal transactivation domain (TA1), DNA binding domain (DBD), oligomerization domain (OD), C-terminal sterile alpha motif (SAM), a second TA domain (TA2), and a transcription inhibitory domain (TID) (Figure 1A) [22,23]. The DBD facilitates protein–DNA interactions, and the TA domain recruits transcriptional coregulators. The TA is also auto-inhibited by the TID. The SAM domain is thought to mediate additional protein-protein and lipid-protein interactions [24]. The DBD and OD domains are shared among all the p63 isoforms, while the SAM domain is present only in the α isoforms [25]. Although the ΔNp63 isoforms lack the N-terminal TA domain found in TAp63, the ΔNp63α and ΔNp63β variants contain a TA2 domain between the OD and SAM domain and has been proposed to possess an intrinsic transactivation activity that is crucial for protein degradation functions to self-regulate protein levels [26,27]. The ΔNp63α isoform is relatively more important for epithelial tissue maintenance owing to its predominant expression [11]. p63 and p53 both share the same general conserved structure; the TA domain, DBD, and OD domains of p63 share 22, 60, and 38 percent identity with those of p53, and the residues p63 uses to interact with DNA are the same in p53, which allows p63 to activate transcription at p53-regulated promoters [28]. In addition, the TAp63 and ΔNp63 isoforms counterbalance the functions of each other in some contexts [29,30]. For instance, while the TAp63α isoform represses cell proliferation, the ΔNp63α isoform promotes cell proliferation and survival [31]. However, the molecular complexity of *TP63* makes it challenging to dissect the specific mechanisms involved in each pathway.

Intramolecular interactions between p53, p63, and p73 have been extensively studied for their overlapping roles in development, cancer, and epidermal homeostasis [32]. p63 and p73 colocalize in the basal epithelium of organs and undergo hetero-oligomerization to form a hetero-tetramer with two molecules of p63 and p73 each [33,34]. p73 has been identified to interact with the ΔNp63α isoform via direct promoter binding, resulting in the negative regulation of p73-mediated apoptosis [35]. It has also been reported that p53 interactions with p63 and p73 result in p53 gain of oncogenic function by suppressing the pro-apoptotic functions of p63 and p73 [36].

**Figure 1 cells-14-00043-f001:**
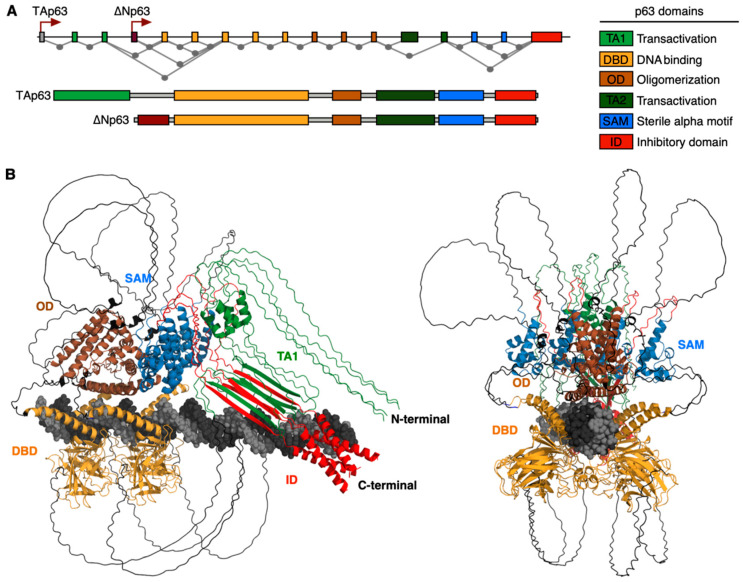
Molecular structure of p63 and DNA binding. (**A**) Structure of the *TP63* gene. Exons are depicted as color boxes. Alternative promoters (arrows) drive the expression of each main isoform, and alternative splicing events are shown. The domain structure of each main isoform is also shown. (**B**) The molecular structure of TAp63 isoform in tetrameric form was predicted based on the amino acid sequence and the p63 binding DNA motif. Four copies of the full-length TAp63 isoform and 2 tandem repeats of the p63 consensus binding motif [TTGGG**CATG**TCCGGA**CATG**CCCAT] (Riege et al., 2020) were used as input for prediction using AlphaFold3 [37]. Color code matches the domains shown in (**A**). Molecular structure file is provided in the Appendix A.

Distinct challenges have restricted the dissection of the molecular structure of p63: the large size of the protein, multiple isoforms, distinct functional domains with different protein conformations, the establishment of tetrameric complexes, regulatory interactions with a wide range of co-factors, and the presence of multiple unstructured domains with intrinsic flexibility. Thus, we exploited AlphaFold 3 [37] to predict the p63 protein structure and its interactions with the target DNA sequences (Figure 1B). To do so, we used the full-length p63 protein sequence and the consensus binding motif as input. We also included the tetrameric complex of p63 in the model and visualized the structure using the PyMOL Molecular Graphics System. In agreement with the experimental evidence, the model predicts interactions between the C-terminal transactivation domain (TA1) and the inhibitory domain (ID), explaining the inhibition mechanism (Figure 1B). The DNA binding domain (DBD) is highly conserved with that of p53 and includes alpha-helices that interact with the major groove of the DNA binding motifs (Figure 1B). Oligomerization and SAM domains from distinct monomers also form alpha-helices that interact with each other to stabilize the tetrameric complex (Figure 1B). Finally, disordered domains connect these structures (Figure 1B). Further structural studies will be needed to validate the p63 structure and the specific domains required for gene regulation. In addition, current advances in next-generation sequencing and long-read sequencing will play an essential role in understanding the specific expression patterns of each isoform in distinct contexts, tissues, and normal and disease states.

## 3. *TP63* and Developmental Control Mechanisms

### 3.1. Germline Mutations of TP63 and Developmental Defects

Dysregulation of p63 results in a wide spectrum of developmental disorders. In fact, hundreds of mutations have been identified and linked to specific diseases (Figure 2B). Germline mutations of *TP63* are the underlying cause of multiple developmental defects, including ectodermal dysplasia, limb malformation, and orofacial clefting [10]. Ectodermal dysplasia can be defined as the malformation of the ectodermal layer of the embryo during development. These defects manifest in various disorders, but the most significant ones are ectrodactyly, ectodermal dysplasia, cleft lip/palate (EEC), ankyloblepharon–ectodermal dysplasia clefting (AEC), acro–dermato-ungual–lacrimal–tooth (ADULT), limb–mammary (LM) and premature ovarian insufficiency (POI). Whether these consequences result from impaired cell lineage specification and/or failure of proper senescence induction pathways remains to be explored. EEC and AEC are autosomal dominant genetic, developmental disorders caused by mutations in *TP63*. They are characterized by abnormal development of ectodermal tissues such as the skin, hair, teeth, eyes, and sweat glands, as well as, in the case of EEC, absent fingers or toes (ectrodactyly) [11,38]. Multiple missense mutations of p63 have been associated with EEC, but the R304W mutation is the most well-characterized and targeted repression of the mutant protein with small interfering RNA (siRNA) can partially rescue p63 function in cells from ECC patients [39]. However, multiple other *TP63* missense mutations have been identified in EEC. These mutations typically alter the transcriptional activity of *TP63* in EEC syndrome and frequently occur in the regions coding for arginine residues of the DNA binding domain (Figure 2B). The AEC syndrome was discovered by Hay and Wells [40] and was found to be a rare heritable autosomal dominant condition; only 100 cases have been reported [11]. AEC is characterized by symptoms such as adhesion of the superior and inferior eyelids (ankyloblepheron), defective formation of the roof of the mouth and lip (cleft palate), and other severe skin defects such as sparse, frizzy hair with small areas of alopecia, erosive dermatitis, and recurrent scalp infections [41,42]. Although initial reports associate AEC with mutations in the C-terminal domain of both TAp63 and ΔNp63 isoforms [43,44], the latest evidence links multiple mutations across the entire gene body with the disease (Figure 1). As an example, the ΔNp63α isoform carrying an L514F mutation fails to interact with the RNA processing complex via the SRA4 protein, hindering RNA polymerase II interactions resulting in altered keratinocyte differentiation and survival while also contributing to AEC [44]. This observation of reduced transcriptional activity of keratinocyte-specific gene promoters was later corroborated by Browne et al. while also demonstrating that the ΔNp63α isoform accumulates in the skin tissue of EEC and AEC patients [45]. Identifying EEC and AEC syndromes using this abnormal protein aggregation as a biomarker opens the door for diagnostics [43].

The ADULT syndrome is also caused by *TP63* mutations in the germline [46]. It is usually identified by symptoms of patients having skin, teeth, and nail defects [10]. Interestingly, among the identified *TP63* mutations that cause ADULT, R298Q results in a gain of transactivation activity of the ΔNp63γ isoform rather than a loss of DNA binding ability, although the mutation is present in the DNA binding domain [47,48,49]. Additional mutations, such as R277Q, also occur in the DNA binding domain [50], P127L [51] or other protein domains, such as G173V [52] and N6H [46]. However, is unclear what are their effects on p63 DNA binding or transactivation activity. Thus, further studies are necessary to understand the underlying causes of the disease pathology.

The main difference between ADULT and the EEC and AEC syndromes is the absence of facial clefting in the ADULT phenotype [11,53]. While EEC and AEC syndromes result from a dominant-negative mutation leading to the inability of p63 to maintain ectodermal development, ADULT syndrome results from transactivation activity gain-of-function of the mutated p63. The second transactivation domain of p63, which is usually repressed in non-mutated isoforms, is activated due to DBD domain mutations resulting in transcriptional activation of the isoform [48,49]. How these mutations located in different regions within the protein result in similar phenotypes remains unclear.

LM syndrome is an autosomal dominant heritable disorder similar to the EEC phenotype and associated with mutations in *TP63* [54]. It manifests as ectrodactyly, lacrimal duct stenosis, hypoplasia of the mammary gland and the nipples, nail hypoplasia and internal female genital dysgenesis, and cleft palate with or without cleft lip [55]. Similar limb malformations occur in LMS as they do in EEC syndrome. From a clinical perspective, the ADULT syndrome overlaps with the EEC and LM syndromes. Originally, LMS was mapped to mutations on chromosome 3q27, particularly a point mutation in the coding regions of the putative second transactivation domain (G76W) of p63 [49,56]. As previously stated, this second transactivation domain is thought to confer transcriptional activation activity to ΔNp63 isoforms of p63. However, recent studies have identified LMS mutations throughout the gene body. Guazzarotti et al. recently identified a novel Y3X stop codon mutation of the ΔNp63 isoform by studying a family with recurring LM syndromes [55]. Furthermore, an R319H point mutation was identified in *TP63* by peripheral blood mutation analysis in a 13-year-old Japanese patient diagnosed with ELA syndromes (EEC/LM/ADULT) [57].

Alterations in *TP63* regulation are also linked to premature ovarian insufficiency (POI). POI is typically characterized by a decline in fertility, menstrual disturbance, and high levels of follicle-stimulating hormone (FSH) before 40 years of age. The TAp63α isoform is expressed in oocytes within the primary follicle and, under physiological conditions, forms an inactive dimer structure [15,58]. As a response to DNA damage, the TAp63α isoform is phosphorylated, resulting in the formation of an active tetramer structure that induces cell apoptosis by upregulating pro-apoptotic factors such as *Puma* and *Bax*, thereby playing important roles in genome quality surveillance of oocytes [59,60]. One study generated the specific POI human mutation in mice (p63^+/R647C^) and identified that mutations in TAp63 cause the formation of the apoptosis-inducing active tetramer structure even in the absence of DNA damage resulting in oocyte loss [59]. Different forms of POI, such as isolated and syndromic POI, resulting from heterozygous *TP63* pathogenic variants have been identified and characterized [61]. These p63 isoform distinctions help us understand the etiology of POI and the differences within the POI spectrum. Importantly, studies also underscore the critical role of both isoforms in monitoring female ovaries and maintaining the genomic integrity of the oocytes. However, the exact mechanism by which the ΔNp63α isoform exerts its function remains unclear.

**Figure 2 cells-14-00043-f002:**
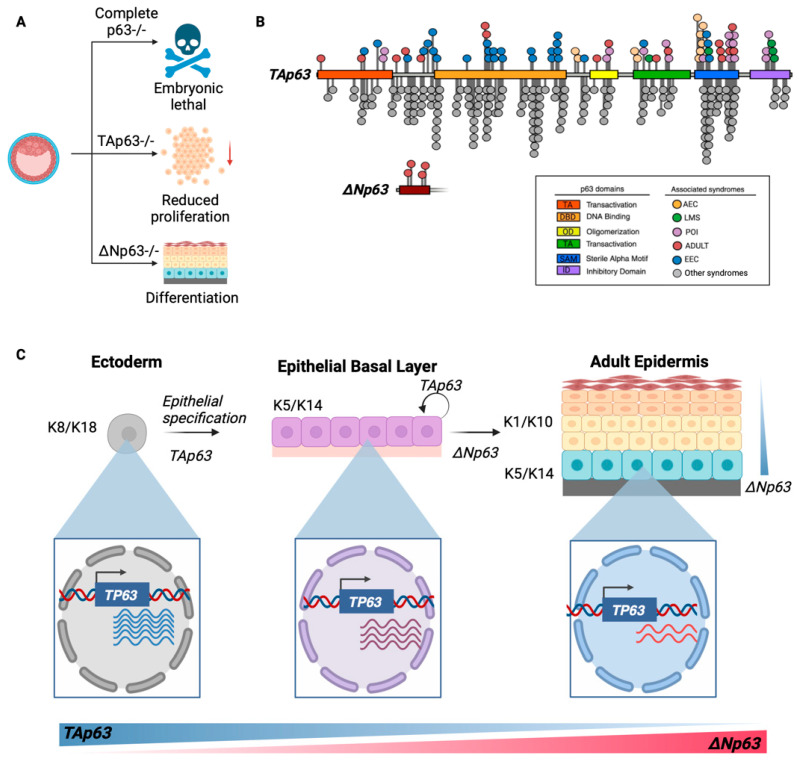
The role of *TP63* in the regulation of developmental control mechanisms. (**A**) Developmental defects of p63 depletion. Complete p63 knockout is lethal in mice models, while targeted depletions of each of the main isoforms result in wide abnormalities across tissues characterized by reduced proliferation and incomplete differentiation. (**B**) Annotated mutations with disease links of the *TP63* gene. Pathological variants were obtained from the ClinVar database [62] and plotted using a lollipop diagram generator [63]. (**C**) The role of p63 in normal epidermis development. TAp63 expression induces epithelial specification in K8/K18 ectoderm progenitor cells to form a basal epithelial layer of K5/K14-expressing cells in which TAp63 acts to promote proliferation. After establishing the basal layer, a switch in p63 isoform expression (from TAp63 to ∆Np63) is triggered to induce epithelial stratification of the epidermis. Panels A and C were made in Biorender.

### 3.2. Regulation of Developmental Mechanisms

*TP63* is a master regulator of cell differentiation and plays a vital role in developmental control [64,65,66]. However, little is known about the specific mechanism of developmental control by p63 during organogenesis. One of the cell lineages better characterized is epithelial development. Studies have shown that ablating TAp63 in precursor basal epithelial cells expressing the marker K14 leads to impaired proliferation, suggesting TAp63 promotes proliferation in these cells. After epithelial specification, these cells differentiate to form the epithelial basal layer and TAp63 is required for stem cell progenitors’ maintenance [67]. Once the epithelial basal layer is formed, ΔNp63 expression is induced to trigger the stratification of the epithelium generating an intermediate filament network with de novo induction of the K5 and K14 basal keratins, and its expression is maintained in the basal layer of stratified epithelial tissue [68]. p63 has also been identified to interact with the cytoprotective transcription factor NRF2 and combinatorially bind to the enhancers and promoters of epidermal genes in keratinocytes. This interaction promotes keratinocyte proliferation in a cyclin kinase 12-dependent manner [69]. Further, the transcriptional termination of a specific subset of early epidermal genes is also coregulated by ΔNp63 via *SETX* interactions [70]. Moreover, a comprehensive analysis across human tissues revealed p63 expression in the basal layers of epithelium, particularly in the vagina, cervix, esophagus, prostate gland, salivary gland, and mammary gland [71].

Evidence from developmental diseases in humans revealed that p63 is essential during embryogenesis for craniofacial, limb, and epidermal development through signaling cascades such as Notch, Hedgehog, and Wnt [72]. ΔNp63α stimulates keratinocyte differentiation by upregulating the transcription of the negative Hh regulator suppressor of fused (SUFU) [73]. Furthermore, the Wnt co-transcription factor LBH downregulates TAp63 expression while upregulating ΔNp63 expression. This is possibly due to the putative β-catenin response elements in the ΔNp63 promoter [74]. As for Notch signaling, the ΔNp63α isoform represses Notch1, thereby inhibiting Notch1-mediated activation of p21 and 14-3-3, which are factors that restrict cell growth to promote differentiation of the keratinocytes [75,76].

Distinct animal models have been developed to characterize the role of p63 in developmental control. In zebrafish, p63 plays multiple functions during gastrulation, including promoting epidermal specification by inhibiting enhancer-Sox3 interactions in neural plate border cells. One of the mechanisms is the binding of p63 to heterochromatin regions to increase accessibility and activate epidermal transcriptional programs [77]. p63 knockout zebrafish models exhibit affected development of epidermal appendages. Researchers also discovered that the TAp63 isoform interacts with p53 to promote keratinocyte proliferation and terminal differentiation by ensuring Notch signaling and caspase 3 homeostasis in zebrafish [78]. In mouse development, p63 controls developmental speed, embryo size, and morphogenesis processes. p63 knockout mice embryos exhibited an approximately 22 h delay in development accompanied by dysmorphology and reduced embryo sizes, resulting in an overall developmental delay [79]. p63 knockout mice models also exhibit affected urothelial development, resulting in columnar epithelial cells in the lower genital tracts destined for squamous and urothelial differentiation [80]. Other mice knockout models have also demonstrated that p63 is crucial for developing the palate and limbs [81]. These studies suggest that all epitheliums require p63 for proper function and development (Figure 2A). While all is known about the expression patterns of p63 in different tissues, the precise details of the mechanisms by which it exerts its influence are yet to be determined. Recently, we found that p63 is induced transitorily during the differentiation of human embryonic stem cells (hESCs) towards multiple lineages, including hepatic, pancreas, mesothelial, muscle, and mesenchymal cells [82]. However, further studies are necessary to better understand the differentiation pathways triggered by p63 and its specific isoforms.

Additionally, p63 is highly conserved among several species, including rats [83], mice [84], humans, and zebrafish [85,86]. Additionally, per the Rat Genome Database [87], p63 orthologs have also been identified in dogs [88], Xenopus [89], squirrels, and pigs. While studies have not demonstrated direct functional replacement between species, the strong conservation of p63 among different species strongly suggests that it plays similar roles in different organisms.

## 4. *TP63* and Adult Tissue Homeostasis

In adult tissues, p63 plays a vital role in the maintenance of epithelial cells in tissues such as the epidermis, respiratory tract, esophagus, bladder, prostate, ovaries, and glandular tissues such as the mammary and thymus glands (Figure 3A,B). ΔNp63 acts to promote epithelial survival, maintain stemness, regulate morphogenesis, and drive differentiation in the epidermis and glandular epithelia and is thus indispensable to the maintenance of adult epithelial tissue [90]. Thymic lineage-restricted p63-knockout mice fail to develop glandular tissues or stratified epithelia, forming only a simple epithelium lacking the spinosum, granulosum, and stratum corneum, underscoring the importance of p63 for proper epithelial development. When fully developed, the epidermis is stratified with a basal layer containing stem cells capable of self-renewal and their daughter transient amplifying cells [2]. ΔNp63 expression is expressed in this basal layer and is necessary for the basal stem cells to maintain the capacity for self-renewal and for keratinocytes to maintain their proliferative activity [68,91].

One way that ∆Np63 acts to maintain this self-renewal capacity in stem cells is by associating with epigenetic coregulators. For example, p63 recruits the DNA methyltransferase 3A (DNMT3a) to active enhancers in epidermal stem cells. It cooperates with Tet2 to maintain high hydroxymethylation levels at the center of DNMT3a-regulated enhancers, thereby regulating epithelial stem cell self-renewal. However, it is not entirely clear if p63 also recruits Tet2 to these sites of active enhancers [92]. p63 also maintains the regenerative capacity of stem cells by regulating cell adhesion molecules. Specifically, in epithelial stem cells, TAp63γ and ΔNp63α have been shown to prevent anoikis, a specific type of apoptosis in anchorage-dependent cells, by upregulating the expression of integrins subunit (*ITGA3*). However, in keratinocytes, ΔNp63α promotes anoikis by upregulating β4 integrin to regulate the adhesion program, leading to detachment of the cells from the underlying mesenchyme. The complex expression thresholds of the TAp63 and the ΔNp63 isoforms determine mechanisms of hyperproliferation or cellular senescence. Whether the adhesion program regulates p63 function and expression in the epidermis or vice versa to regulate epithelial tissue maintenance and survival is not fully understood [93,94].

Glandular epithelial structures also rely on the activity of p63 for proper maintenance, with notable examples being the mammary, prostate, and thymus glands. The mammary epithelium comprises two main cell types, each regulated by distinct isoforms of p63: an inner luminal layer and an outer basal layer containing stem cells [95] (Figure 3C). The TAp63α isoform is highly expressed within the luminal cells and controls cell proliferation, while ΔNp63α is mainly expressed in the basal progenitor cells and promotes quiescence through the regulation of Notch3 [91,96]. Within the mammary gland, it has been shown that ΔNp63 governs morphogenesis, and ablating *TP63* results in a defective phenotype of the mammary gland [97]. Furthermore, germline mutations in *TP63* result in mammary hypoplasia in humans [56]. Additionally, it has been shown that ΔNp63 maintains the self-renewal capacity of mammary stem cells by upregulating the expression of the Wnt signaling receptor Fzd7 [74]. The ΔNp63α isoform also helps to maintain the mammary glands through changes in physiological stages such as puberty, pregnancy, lactation, and post-lactation [98].

Like mammary gland development, mice lacking p63 also fail to form a prostate bud, highlighting the necessity of p63 for proper prostate development [91,99]. Three major cell types comprise the prostate epithelium: basal cells consisting of stem and TA cells, luminal cells, and neuroendocrine cells [100] (Figure 3D). ΔNp63α is expressed only in the basal stem cells, where it is responsible for maintaining their self-renewal capabilities [91,101]. Using an in vitro prostasphere formation assay, one group showed that inhibiting p63 expression decreased the number of prostate stem cells and increased the number of differentiated luminal cells expressing the differentiation marker CK8, which underscores the importance of p63 for maintaining epithelial stem cell populations in the prostate [102].

As with the mammary and prostate glands, p63 is indispensable for proper thymus development. Mice with a p63-knockout in thymic epithelial stem cells display severe thymic dysplasia characterized by a poorly defined separation of medullary and cortical compartments [103]. Within these compartments, ΔNp63 is the dominant isoform found in thymic epithelial stem cells, and it has been shown that the expression of ΔNp63α within the pharyngeal pouch is crucial for maintaining these stem cells [91]. Interestingly, while no skin defects were observed in mice with a p63 knockout in thymic epithelial stem cells, all adult mice lacked hair follicles [103].

The maintenance of other epithelial appendages, such as hair, is also regulated by p63 activity [104,105]. Stem cell populations located within several niches of the hair follicle act to maintain the epithelium through the activity of ΔNp63α, which is expressed only in these stem cells as well as the outer root sheath and matrix cells [105]. The hair follicle stem cell niche was depleted in mice with mutant ΔNp63 [106]. Additionally, mice with p63 knockout produce progeny without hair follicles, illustrating the critical role p63 plays in developing epithelial appendages [97,107].

The lung is lined with a ciliated pseudostratified epithelium that acts as a barrier between the internal lung tissue and the outside environment (Figure 3E). Progenitor cells called club cells within the lung produce epithelial alveolar cells and respond to lung damage [2]. ΔNp63 is the primary isoform found in the lung and is exclusively expressed in the basal cells of the bronchial epithelium [108]. Previous work has shown that p63-expressing distal airway stem cells undergo proliferation in response to lung damage [109,110]. Furthermore, one group investigating the role of the mesenchymal stem cell secretome in acute lung injury recently found that p63 promotes proliferation in epithelial progenitor cells to repair damaged cells, demonstrating the vital role p63 plays in the maintenance of lung epithelial tissue in response to lung damage [111]. Primary bronchial epithelial cells with a p63 knockdown showed signs of senescence and decreased proliferation, highlighting the importance of p63 activity in maintaining the proliferative capability of epithelial cells in the lung [108].

In summary, p63 is a major regulator of the growth, maintenance, and differentiation of epithelial tissue such as the epidermis, glandular epithelia, skin appendages, and lung lining. The activity of the primary isoform found in these tissues, ΔNp63, is especially critical for basal stem cells to maintain stemness and their self-renewal capabilities. A diverse array of miRNA molecules and other proteins, such as epigenetic coregulators and proteins involved in the Hedgehog, Notch, and Wnt signaling cascades, act with p63 to exert its effects and modulate its activity.

## 5. Role of p63 in Genome Organization and Function

The human genome is organized in functional compartments and chromosomes fold into specific patterns to enable proper gene regulation [112]. This genome compartmentalization aligns with the temporal order of DNA replication (replication timing), with active compartments replicating early during S-phase and inactive compartments replicating later [113,114]. Replication timing and large-scale chromatin organization also change dynamically during cell differentiation in coordination with transcriptional activity [115,116]. However, control mechanisms of replication timing and genome organization remain elusive. An emerging hypothesis is that pioneer transcription factors can shape the epigenome to establish the chromatin contexts required for transcriptional induction during cell differentiation [117,118,119,120]. Recent findings suggest that p63 is a key master regulator of genome organization and chromatin landscape.

The p63 transcription factor is required for the activation of multiple genes and regulatory elements. The mechanism of p63 regulation includes the remodeling of the chromatin and the establishment of regulatory chromatin interactions that can trigger cell fate differentiation decisions or even induce cell transdifferentiation. In fact, p63 can directly recruit remodeling complexes such as SWI/BAF, work as pioneer transcription factor by reposition nucleosomes and promote gene expression [121,122,123]. In addition, p63 also interacts with histone modification enzymes (such as p300) to promote chromatin accessibility; however, effects on gene expression are context dependent [124,125,126]. Consistent with these observations, p63 overexpression in cancer cells (see cancer section below) can reprogram the enhancer landscape and trigger transdifferentiation [127]. One of the most dramatic examples of p63 function in chromatin remodeling is during epidermal development. During this process, the keratinocyte lineage-specific gene locus, epidermal differentiation complex (EDC), undergoes complex rearrangements in the nucleus. In early progenitors EDC locus is situated in the nuclear periphery within inactive chromatin domains; however, it translocates to the nuclear interior during differentiation to an active region highly enriched in SC35-positive nuclear speckles (Figure 4A) [128]. Using p63 knockout mouse models, Fessing et al. demonstrated that p63 ablation results in lower expression of chromatin remodeling factors such as *Satb1* and *Brg1*, resulting in altered epidermal morphology during keratinocyte maturation [129]. *Brg1*^−/−^ mouse models confirmed the importance of this interaction for internalization of the EDC locus [128]. In addition, p63’s interactions with the catalytic subunits *Brg1/Brm* of the chromatin remodeler BAF complex are crucial to maintaining open chromatin conformations during epidermal specification to improve p63 binding site accessibility [130]. Another factor directly influenced during p63-regulated epidermal differentiation is the Polycomb group member, *Cbx4*, which is pivotal for maintaining epidermal identity during development by repressing specification towards non-epidermal lineages. Using ChIP-seq and transcriptome data from epidermal cells from knockout embryos, Mardaryev et al. identified that *Cbx4* expression depends on p63 [131]. p63 also influences nuclear morphology and chromatin organization via directly binding to nuclear envelope-associated proteins such as Sun1, Plec, and *Syne3*. In p63 knockout mouse models, epidermal-specific reduction in nuclear lamin and other nuclear envelope proteins resulted in the global reduction of known heterochromatin marks—H3K9me3, H3K27me3, and H2AK119Ub [131,132]. Using ATAC-seq, Qu et al. also found differential chromatin accessibility between wild-type and p63 mutant EEC samples enriched at CTCF binding motifs [133]. Additional findings suggest that p63 interacts with the transcription factor TFAP2C and morphogens such as retinoic acid and BMP to prime the chromatin landscape and alter chromatin accessibility [134,135]. Moreover, p63 binding is also recognized by active histone modifications such as H3K27ac or H3K4me1 [136] and recruits DNA methyltransferases [92].

p63 also interacts with histone acetylases (HATs) and deacetylases (HDACs). Specifically, the ΔNp63α isoform interacts with the HAT p300 to form a coactivator complex for β-catenin activation [126]. This isoform interacts with nucleosomes and binds to inaccessible and unmodified chromatin regions to promote acetylation marks. It also enables nucleosome repositioning to allow chromatin accessibility [137]. Interactions of p63 with HDAC 1 and 2 are also critical in epidermal stratification and hair follicle specification. Specifically, the interaction between HDAC1/2 and ΔNp63α is crucial for the co-repression of cell cycle inhibitory proteins *p16/Ink4a* and *p19/Arf* [138]. Moreover, in cancer cells, the ΔNp63 isoform is recruited to target genes to promote chromatin accessibility and gene expression for cancer proliferation, metastasis, and migration [139,140]. These studies suggest a role for p63 in orchestrating chromatin integrity and genome organization. Integrating these observations, a model for p63’s regulation of genome organization emerges (Figure 4B). During differentiation, p63 can bind to the nucleosomes of inaccessible heterochromatin regions to recruit p300 and induce histone acetylation. It could also enable the establishment of the epidermal enhancer landscape by aiding the formation of the enhancer-promoter loop via CTCF interactions and the recruitment of transcriptional machinery. In this case, these changes are accompanied by subnuclear reorganization and increased transcriptional activity of the EDC locus. These findings present an intriguing possibility of p63’s pioneering role in regulating nuclear morphology, epigenome, and genome organization in epidermal development. Further research on this matter would shed more light on the mechanistic nuances of p63 function and open avenues for a better understanding of epidermal development and diseases.

## 6. Roles in Senescence and Aging

Senescence is characterized by cell growth arrest and can be triggered by several intrinsic and extrinsic factors, such as the accumulation of reactive oxygen species (ROS) or UV radiation [141]. Previous work has shown that p63 is involved in regulating cellular senescence in keratinocytes [142]. However, due to the complexity of this gene, the dissection of the specific pathways and roles of each isoform have been challenging. In vitro studies have found induction of replicative senescence in human fibroblasts by p63 via direct interactions with *cdk1* and *cyclin B* genes to inactivate the NF-Y transcription factor [143]. However, p63-deficient mice express increased senescence markers such as SA-β-Gal, PML, and p16^INK4a,^ suggesting distinct mechanisms between p63 and senescence response [144]. For example, TAp63-knockout in mice results in epidermal stem cell senescence by the downregulating of *INK4A* and *ARF* [145]. However, in human fibroblasts, overexpression of TAp63 promotes senescence by activating the cellular senescence marker p21, illustrating how p63 roles in senescence are cell type-specific [146]. Thus, cell type-specific and isoform-specific analyses are necessary to unveil the roles of p63 in senescence control. Further, p63 effects on senescence control are also extended through interactions with miRNAs. Specifically, miRNAs miR-574-3p and miR-720 have been shown to indirectly increase amounts of ΔNp63 in keratinocytes and accelerate cell differentiation [147]. miR-203 and miR130b are additional miRNAs that can inhibit ΔNp63 to induce cell cycle arrest in epithelial precursors and induce senescence in primary human keratinocytes [142,148,149]. Moreover, ΔNp63α works to regulate senescence in keratinocytes by downregulating expression of the senescence-inducing miRNAs miR-130, miR-138, and miR-181a/b thereby promoting keratinocyte proliferation [68,142]. Consistent with these findings, Keyes et al. have demonstrated that high levels of ΔNp63 inhibit senescence-associated β-galactosidase (SA-β-Gal) activity in keratinocytes to promote hyperproliferation while its downregulation results in oncogene-induced senescence (OIS) [150]. Overall, these studies suggest that a delicate balance in the expression of p63 isoforms can regulate cell proliferation and cellular senescence response.

In addition to senescence and other phenotypic changes that occur during aging, many epigenetic changes have been linked to aging over the last several decades. As previously mentioned, many epigenetic regulators interact with p63, so unsurprisingly, its dysregulation has also been associated with aging-related pathologies. A reduction in DNA methylation in aged fibroblasts from several species was observed in the 1980s, the concept of the “epigenetic clock” describing epigenetic age and linking DNA methylation levels to age was proposed in 2013, and recent single-cell profiling techniques provide new insight into the relationship between epigenetics and aging [151,152,153]. Interestingly, one group recently demonstrated a link between p63 and epigenetic age by showing that tissue fractions from primary neonatal skin cells enriched in stem cells were epigenetically younger in terms of DNA methylation and expressed significantly more p63 compared to a stem cell-depleted fraction [154]. The activity of p63 has also been linked to epigenetic coregulators such as the histone methyltransferase 2B (*KMT2B*), DNA methyltransferase 3A (*DNMT3A*), histone acetylase p300, and Class III Histone Deacetylase (*HDAC*) or Sirtuins (*SIRTS 1-7*) proteins. *KMT2B* interacts with enhancer-bound ΔNp63 at epidermal target genes to deposit histone 3 lysine 4 monomethylation marks (H3K4me1) to prepare enhancers for activation. The absence of *KMT2D* hinders p300 recruitment to the enhancers and prevents transcriptional activation of key epidermal genes, resulting in dysregulated epidermal stratification [155]. As previously discussed, *DNMT3A* is recruited to p63-bound active enhancers via H3K36me3 in epidermal stem cells to regulate epidermal differentiation. *DNMT3A* physically associates with p63 at these active enhancers to maintain high DNA hydroxymethylation (5-hmC) levels at the enhancer core [92]. Moreover, ΔNp63α also interacts with p300 in vivo, catalyzing the acetylation of lysine 193 (K193ac) and stabilizing p63. ΔNp63 was also found to inhibit the expression of SIRT1-inhibiting senescence-inducing miRNAs, and *SIRT1* expression is decreased during replicative senescence in human dermal fibroblasts. While direct *SIRT1* and p63 interactions have not been demonstrated, this finding suggests a possible interaction between p63 and *SIRT1* in keratinocyte senescence [142,156].

Consistent with the role of p63 in senescence induction, *TAp63* knockout mice had decreased life spans and exhibited premature aging signs such as kyphosis, skin ulcerations, reduced hair growth, senescence of epidermal stem cells, and genomic instability, demonstrating a link between TAp63 and premature aging [145]. p63-deficient mice exhibit signs of early aging, such as alopecia and skin defects [144]. Moreover, p63 has been implicated in a particular group of premature aging disorders termed progeroid syndromes, which are a group of rare genetic disorders characterized by accelerated aging [82,157]. Loss of epidermal stem cells in progeria mice models was found with concomitant downregulation of ΔNp63, leading the group to hypothesize that downregulation of p63 caused the observed loss of stem cells and decreased proliferative potential in progeria cells [157]. Moreover, we recently identified altered regulation of *TP63* in progeroid diseases with independent causal mutations [82,158]. In fact, alterations in *TP63* regulation were detected in cells derived from patients with HGPS, Rothmund-Thomson, and Werner syndromes, as well as healthy aged donors [82,158]. These alterations are associated to an imbalanced p63 isoform expression with higher levels of TAp63 [82,158]. Furthermore, p63 expression alterations have been also found in other premature aging syndromes such as xeroderma pigmentosum (XP), Cockayne syndrome (CS), trichothiodystrophy (TTD), premature menopause, and age-related lung disease. Previous work has shown that p63 expression was decreased in primary keratinocytes from patients with XP, CS, and TTD compared to normal human keratinocytes [159]. Similarly, p63 is involved in lung aging and other age-related lung diseases. One group found that while ΔNp63 expression was upregulated in an aged mice group compared to a younger cohort, nuclear TAp63 was reduced in both aged mice and the rhesus monkey group [160]. These findings show that mutation or dysregulation of *TP63* and its gene products can contribute to and cause age-related pathology in epithelial tissue.

These studies highlight the key roles both major isoforms of p63 play in regulating senescence via modulating the expression of target genes encoding cell cycle-related proteins or senescence-inducing miRNAs or through interactions with epigenetic regulators. Furthermore, the work described herein demonstrates a link between p63 and aging and shows how p63 dysregulation can lead to age-related pathologies and premature aging diseases.

## 7. Role in Cancer

Altered expression of the *TP63* is linked to cancer progression and metastasis. p63-p53 interactions have been associated with the activation of tumor suppression cascades [161,162]. Since the two major isoforms are expressed at different levels across cell types and are driven by non-overlapping transcriptional networks, this distinct expression pattern might help facilitate the dissection of their roles [163]. Furthermore, interactions with p53 and p73 establish complex regulatory pathways still under investigation. Nevertheless, here we discuss the latest advances in the role of p63 in distinct types of cancers.

### 7.1. Lung Adenocarcinoma and Lung Squamous Cell Carcinoma

Genome-wide association studies (GWAS) have identified *TP63* as a top-risk locus for lung adenocarcinoma (LUAD). *TP63* single nucleotide polymorphisms (SNPs), amplification, and gene overexpression have been observed in LUAD patient samples [164]. The TAp63-specific SNP rs488809, along with other SNPs such as rs10937405 and rs439680, were all found to be associated with LUAD in various populations [165,166,167,168]. ΔNp63 functions as a pivotal oncogenic driver in LUAD and LUSC by maintaining distal lung stem cell proliferation and regulating basal cell identity gene expression such as BCL9L by cooperatively altering the chromatin landscape with the histone acetylase p300 at enhancer-associated genes to promote the expression of these genes. Its dual role in driving tumor initiation and supporting the cellular architecture is essential for tumor growth, positioning ΔNp63 as a vital mediator of LUAD and LUSC tumorigenesis [169]. Taken together, these results suggest that ΔNp63 regulates the expression of its target genes by altering the chromatin landscape to promote tumorigenesis and cancer progression in LUAD and LUSC [169].

### 7.2. Squamous Cell Carcinoma

*TP63* is often implicated in epithelial cell cancers such as squamous cell carcinoma (SCC), lung, and breast cancer. Overexpression of *TP63* has been observed in SCCs of the head and neck, esophagus, lungs, etc. [170,171,172]. In the study by Ramsey et al., the knockout of *TP63* in mice with advanced SCC led to tumor regression, indicating that *TP63* may have an oncogenic role under these conditions, similar to LUAD/LUSC [173]. Conversely, Lakshmanachetty et al. reported that the knockdown of *TP63* before tumor induction led to aggressive tumor development, suggesting that *TP63* may play a protective role during early tumorigenesis [172]. This discrepancy highlights the importance of experimental design, as the timing of *TP63* knockout appears to influence its functional role in SCC progression significantly.

Separate studies have identified other key signaling axes involving p63 in SCC, and these signaling pathways provide new targets for therapy. For example, one group recently found that ΔNp63α negatively regulates Rac1, a “master switch” of cell motility, by decreasing expression of the Rac1 guanine exchange factor (GEF) *PREX1* and consequently decreasing cell invasion of SCC cells. The results from this study contradict those from Ramsey et al., where the loss of ΔNp63α expression resulted in reduced tumor progression, and support the findings of Lakshmanachetty and colleagues, where the loss of ΔNp63α expression would be associated with increased tumor invasion and metastasis. This study ΔNp63α/Rac1 axis as a potential target for treating SCC [174]. These findings suggest that depending on the pathway it regulates, p63 may promote or impede tumor progression in SCC, perhaps subtype-specific.

### 7.3. Bladder Cancer

In bladder cancer, loss of *TP63* expression is associated with cancer progression, which highlights the contrasting roles of p63 and/or its specific isoforms in different contexts and tissues. Although the expression of *TP63* is maintained in superficial bladder tumors, a loss of *TP63* is observed in the more invasive bladder tumors, highlighting that loss of p63 expression promotes cell invasion in bladder cancer [175]. Similarly, another study using patient samples observed a significant loss of *TP63* expression with more aggressive bladder cancers and also found an increase in Ki67 expression, a proliferation marker, suggesting that low *TP63* expression is needed for bladder cancer progression [176]. In meta-genome-wide association studies looking at populations across the globe with and without bladder cancer, the SNP rs710521 located near the *TP63* gene has been strongly associated with an increased risk of bladder cancer, showing that mutations in *TP63* are essential for the development of bladder cancer [177].

In a more recent study looking at specific isoforms of *TP63* and its correlation to the survival of cancer patients, the authors performed an RNA-seq of patient samples and correlated gene expression to patient outcome [178]. The authors found that the TAp63 isoform of the *TP63* gene correlated with worse survival, especially in the basal bladder cancer subtype. Surprisingly, the ΔNp63 isoform, an oncogene in other cancers, correlated with better survival and lower relapse in luminal bladder cancer patients [178]. An opposite trend was observed in skin cutaneous melanoma patients, where ΔNp63 correlated with worse survival [178]. There may be a balance between the expression of ΔNp63 and TAp63 during bladder development, with ΔNp63 playing crucial roles in terminal differentiation. However, in cancer, where there is an increased expression of the TAp63 isoform, the ΔNp63 isoform cannot initiate terminal differentiation, resulting in hyperproliferation or tumor progression in this case [179]. These findings show that the isoforms of p63 have different functions in promoting cancer progression and cell survival in various cell types.

### 7.4. Breast Cancer

As described previously, *TP63* plays an integral role in the early development of the mammary glands, and it is also essential for maintaining mammary stem cells and regulating their differentiation into mammary progenitors and other cell types in the mammary gland [98]. *TP63* may play contradictory roles in breast cancer, which may be due to the differential expression of the two isoforms of *TP63* in different tumors, with the TAp63 isoform being negligibly expressed in the breast epithelial cells while the pro-tumorigenic ΔNp63 is highly expressed in basal-type triple-negative breast cancer [180]. However, a small subset of luminal tumors also exhibits ΔNp63 expression and is associated with aggressive tumors [181]. In most breast cancers, upregulation of the isoform ΔNp63 is observed and generally plays an oncogenic role in this context [170]. Interestingly, Bui et al. have identified that the oscillatory expression of ΔNp63 is crucial for the progression and invasion of breast cancer, suggesting that while ΔNp63 is required for tumor initiation, its loss is necessary for metastasis [182].

Kim et al. found that in estrogen receptor alpha positive (ER alpha +) breast cancer cell lines, *TP63* downregulation by the miRNA Hsa-miR-196a2* decreased the proliferation and invasiveness in the breast cancer cell line MCF-7, which shows that p63 acts to promote cell proliferation and cell invasion in breast cancer [183]. Additionally, IL13Ralpha2 deletion upregulated *TP63* and significantly reduced metastasis of breast cancer cells to the lungs, suggesting that *TP63* acts as a tumor-suppressor gene and prevents metastasis in BLBC [184]. In a more recent study exploring mechanisms of drug resistance in breast cancer cells, it was demonstrated that inflammation in tissues leads to the upregulation of IL-1B and, subsequently, ΔNp63, which upregulates the expression of growth factors and downregulates the expression of DNA damage sensors, leading to drug resistance and the progression of breast cancer [185]. These results contradict those found by Papageorgis et al., where IL13Ralpha2 deletion activates the STAT6-*TP63* pathway and upregulates *TP63* to suppress breast cancer tumors [184]. However, Papageorgis et al. did not perform isoform-specific *TP63* expression and function analyses. Therefore, it is possible that the STAT6-*TP63* pathway may lead to the upregulation of TAp63 and not the ΔNp63 isoform, which would explain the tumor-suppressive effect of its expression [186]. More isoform-specific studies need to be performed to confirm this hypothesis and resolve the discrepancies in the role of *TP63* in breast cancer.

### 7.5. T-Cell Lymphomas

Peripheral T-cell lymphoma (PTCL) is a rare form of cancer that develops from mature T-cells and NK cells with a survival rate of 20–30%. A study by Vasmatzis et al. identified chromosomal rearrangements within *TP63* in 11 (5.8%) out of 190 total PTCL samples. These rearrangements generated truncated versions of p63 homologous to ΔNp63 and were associated with overall decreased survival. The function of these homologous truncated proteins was not interrogated by the authors. However, overexpression of ΔNp63 might be the cause as it is associated with oncogenic functions in other cancers, such as SCC. Thus, increased expression of a truncated protein homologous to ΔNp63 may have similar functions to wild-type ΔNp63 and promote cancer in PTCL. More studies investigating the exact functions of these truncated proteins in the context of PTCL are needed to parse out the exact role p63 may play in decreasing survival and promoting cancer progression in PTCL [187].

Anaplastic large cell lymphomas (ALCLs) represent a group of CD30-positive T-cell lymphomas that often contain chromosomal rearrangements in the *ALK* locus and, similar to PTCLs, may contain *TP63* rearrangements which are generally associated with poor patient outcome [188]. Wang and colleagues found that 35.3% of the 116 ALCL cell lines examined were p63-positive, with p63 being seen more frequently in ALK-negative ALCL. Previous reports have shown that ALCL cases with no *TP63* rearrangements typically contain extra copies of *TP63*, which leads to an imbalance of isoform expression, and indeed, this group found that all ALCL cell lines without chromosomal rearrangements exclusively expressed the TAp63 isoform with zero tested cell lines being positive for ΔNp63 [188]. Of note, this group did not find any significant difference in clinical outcomes or overall survivability of ALCL cells between p63-positive and p63-negative cases, which suggests that the expression and function of p63 are not crucial for pathogenic outcomes or cell survivability in ALCL as they are in other cancers. However, this study illustrates how p63 can be used as a prognostic marker by immunohistochemistry in ALCL samples. Additionally, these studies demonstrate that depending on the type of T-cell lymphoma, TAp63 or ΔNp63 expression may be responsible for the observed cancer phenotype; however, more intensive studies investigating the exact role of each isoform in ALCL are needed to establish a more concrete conclusion on the function of p63 in T-cell lymphomas.

### 7.6. Lymphocytic Leukemia

As previously stated, the apoptotic effects of TAp63 can be dysregulated in many cancers, and deficient apoptosis is a characteristic of chronic lymphocytic leukemia (CLL). TAp63 is the more prevalent isoform found in CLL, and one group found that siRNA-mediated downregulation of TAp63 afforded some protection against spontaneous apoptosis in CLL cell lines and increased cell proliferation in Raji lymphoma cells, which is consistent with previous results that demonstrate the pro-apoptotic function of TAp63 [189]. Mechanistically, the group found that TAp63 is epigenetically silenced by the miRNA mir-21, which is upregulated following B-cell receptor engagement. A 2019 study analyzing the DNA methylation levels of the genome from CLL samples observed hypomethylation and upregulation of *TP63*, and siRNA-mediated downregulation of *TP63* increased apoptosis in CLL cells, which demonstrates that *TP63* has a pro-survival effect on CLL [190]. Similar to Humphries and colleagues, Papakonstatinou et al. identified TAp63 as the most prevalent isoform in CLL; however, TAp63 appears to function in opposite manners in these two studies [189,190]. This discrepancy in the role of *TP63* as pro-apoptotic versus pro-survival in CLL may be attributed to different methodologies used or the different serotyped subtypes of CLL from the patient samples [190].

### 7.7. Pancreatic Carcinoma

Pancreatic carcinomas (PC) represent the third leading cause of cancer death in US cancer patients. Similar to the case in CLL, miRNA-mediated downregulation of *TP63* is associated with cancer progression in PC. One group found that overexpressing the miRNA miR-301b in PC cell lines downregulated *TP63* in five cell lines and promoted cell invasiveness in two of the PC cell lines, suggesting that p63 prevents cell invasion in PC. They found that expressing miR-301b and knocking down *TP63* using shRNA downregulated *CDH1* and activated NF-kB in PC cell lines. Previous work demonstrated the tumor-suppressor function of the *CDH1* gene, and *TP63* has been proposed as genomic target of the oncogene miR-301b, which promotes cell invasiveness in PC through the downregulation of *CDH1* [191,192].

p63 expression can also function to change cell identity in cancer. For example, one separate group demonstrated that ΔNp63 expression controls the transdifferentiation of pancreatic ductal adenocarcinoma (PDA) into a squamous subtype and promotes cell invasiveness through enhancer reprogramming [127]. These results are consistent with those identifying ΔNp63 as an oncogene in cancer. Ectopic expression of ΔNp63 increased the expression of squamous markers KRT5/6 and S100A2 at both the RNA and protein levels, showing that ΔNp63 expression is sufficient to produce squamous-like characteristics in PDA. Previous research demonstrated that lineage changes in PDA occur through alterations of chromatin state. A clustering analysis of ChIP-seq data against H3K27ac and p63 showed that two squamous-like ΔNp63-expressing PDA cultures have a unique enhancer configuration with enrichment of H3K27ac and p63 at enhancers near genes expressed in the squamous subtype of PDA, such as S100A2 or KRT5, suggesting that the enhancer landscape of PDA tumors resembles cells of a squamous cell lineage and is linked to p63 occupancy [127]. Furthermore, SUIT2 cells expressing ΔNp63 in vitro formed more invasive projections, and mice transplanted with SUIT2 cells expressing ΔNp63 formed larger tumors and metastatic lesions compared to the controls, suggesting that ΔNp63 expression is important for cell invasion, tumorigenesis, and metastasis in PDA. Cas9-mediated ΔNp63 inactivation completely arrested tumor growth in mice transplanted with BxPC3-Cas9 cells transduced with *TP63* sgRNAs, suggesting that PDA tumors rely on p63 for tumor growth in vivo. These data indicate that ΔNp63-induced enhancer reprogramming improves the metastatic characteristic and growth potential of PDA tumors [127].

Taken together, the aforementioned studies have implicated TP63 in its clear, complex, and pivotal role in the predisposition, progression, and metastasis of various kinds of cancer, including the role of miRNA interactions.

## 8. Conclusions

In this review, we emphasize the role that p63 plays in developmental regulation, tissue maintenance, senescence, cancer, and genome organization. Our work highlights *TP63* mutations that lead to numerous developmental diseases such as EEC, AEC, ADULT, and LMS syndrome. The existence of these diseases further emphasizes the role that p63 plays in epithelial development. *TP63* is also necessary for certain adult cell types to replenish and function correctly, including skin, hair follicles, prostate gland, mammary gland, thymus, lung, and oocytes. The molecular complexity of p63 activity arises from the existence of its two main isoforms and the various mechanisms by which they trigger distinct pathways in different cellular contexts. TAp63 controls mostly early progenitors, and dysregulation can alter these progenitors and their derivative populations. ∆Np63 is expressed primarily in committed cells and controls proliferation, but overexpression above a certain threshold might induce senescence. The diverse and complex network of interactions in which p63 participates opens up many opportunities to develop therapeutic strategies for treating the various diseases and disorders outlined in this review. The road to implementing treatment for p63-related diseases is long, and ethical considerations constrain its use in humans; however, the work of the studies described here serves as hopeful and promising platforms to launch future research projects. Overall, the recent findings suggest a key role of p63 in regulating a myriad of processes linked to cell differentiation, adult tissue homeostasis, and cancer and aging-associated pathologies. Although multiple challenges exist for dissecting the molecular mechanism of p63 control, the latest advances in genomic approaches that enable the sequencing of full-length transcripts will enable the detection of isoform-specific expression in normal and disease contexts. Advances in stem cell biology and pluripotent cell differentiation systems will also provide excellent tools to dissect the temporal requirements of p63 isoforms during cell fate specification. Finally, novel computational approaches that integrate multimodal signals would play an important role in identifying the complex regulatory interactions driving p63 function.

## Figures and Tables

**Figure 3 cells-14-00043-f003:**
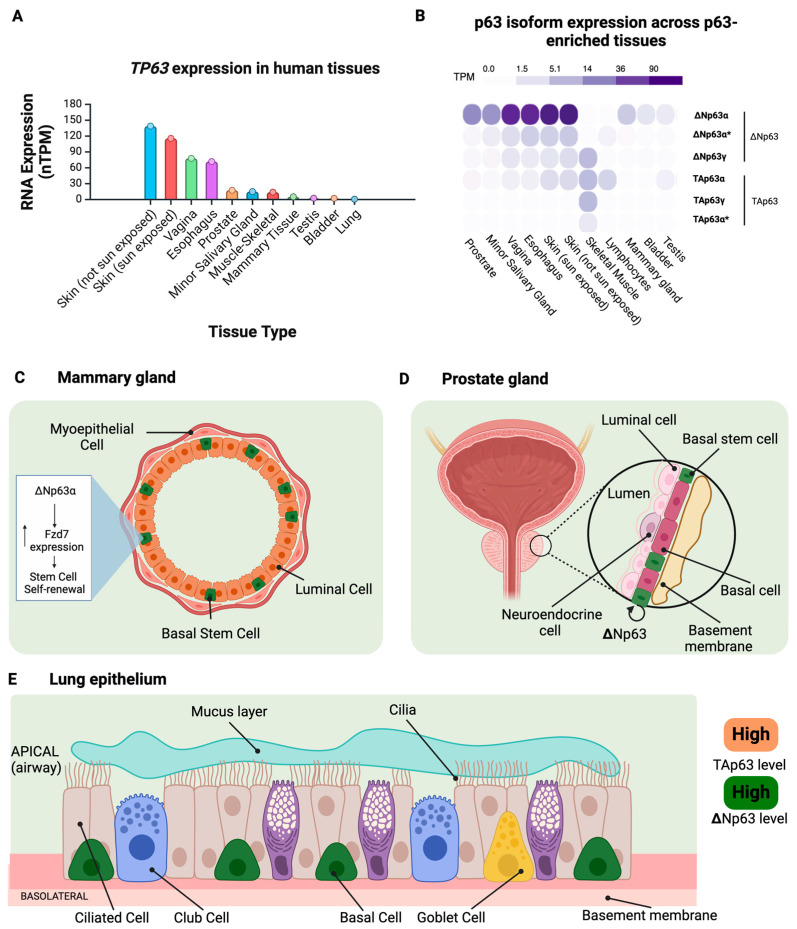
TAp63 and ΔNp63 expression across human adult tissues, glandular tissues, and epitheliums. (**A**) Human adult tissues enriched in *TP63* expression shown as normalized transcript per million (nTPM); (**B**) p63 isoform expression shown as transcripts per million (TPM) in tissues enriched for *TP63* expression. The data used for the analyses described here were obtained from the GTEx Portal on 07/15/24. (**C**) Model showing the cross-section of mammary gland epithelium composed of myoepithelial cells, luminal cells, and basal stem cells enriched for ΔNp63, which upregulates Fzd7 expression to promote stem cell self-renewal. (**D**) Model of human prostate gland displaying luminal, neuroendocrine, and ΔNp63-enriched basal stem cells. (**E**) Schematic illustration of the pseudostratified epithelium of the lung, including ciliated cells, goblet cells, club cells, and basal cells. ΔNp63 is the dominant isoform expressed in the basal cells on the basolateral side of the epithelium. Figure made in Biorender. * *p* < 0.05.

**Figure 4 cells-14-00043-f004:**
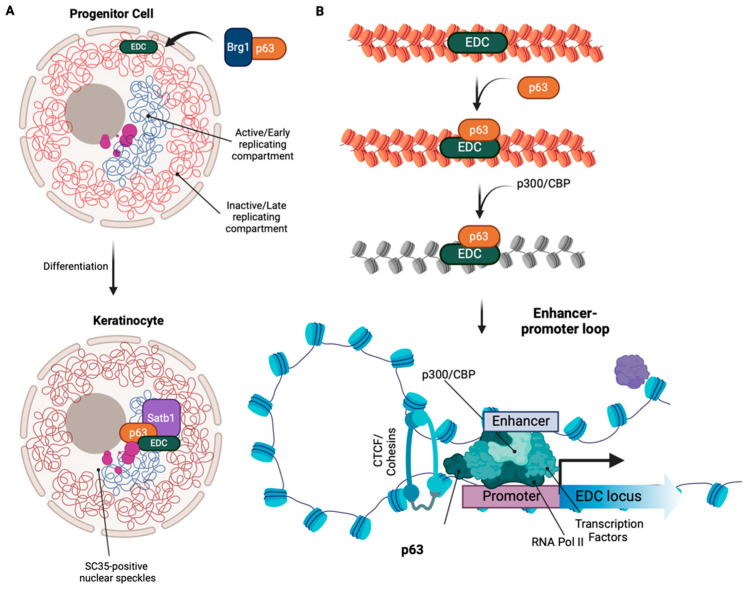
The role of p63 in chromatin remodeling and genome organization. (**A**) p63 recruits Brg1 and Satb1 to mediate the translocation and subsequent expression of the EDC locus near regions of SC35-positive nuclear speckles during keratinocyte differentiation. (**B**) A proposed model in which p63 acts as a pioneer transcription factor to reprogram the EDC locus. p63 binds to compact inactive chromatin, recruits chromatin remodelers to increase accessibility (p300), mediates the formation of the enhancer-promoter loop via the recruitment of crucial transcriptional factors and CTCF, and promotes transcriptional activation of the epidermal genes during keratinocyte differentiation. Figure made in Biorender.

## Data Availability

Not applicable.

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
