# Peer review of "p63: A Master Regulator at the Crossroads Between Development, Senescence, Aging, and Cancer"

_cells, 2025, doi:10.3390/cells14010043_

Round 1
Reviewer 1 Report
Comments and Suggestions for Authors
This is an excellent review article that covers various aspects of p63, including its functions in normal and cancer cells with complexity arising from functionally distinct isoforms, its roles in senescence and aging, its mutation-associated diseases, and its role in chromatin remodeling and organization. The reference list contains recent ones and looks good as well. Minor suggestions are as follows.
· Figure 3 should be cited in the text where appropriate, each separately as 3A, 3B, 3C, 3D and 3E. This comment also applies to some other figures. In terms of referring to the figures in the text, the current version is not reader-friendly.
· Related to line 157: How conserved is p63 among species, e.g., human, mice, zebrafish etc.?
· Line 188, Tap63 to ΔNP63: Uppercase and lowercase letters to be corrected.
· Line 231: R298 to what amino acid?
· Line 286: Figure 2C, not Figure 3?
· Line 478: “Rivera-Mulia, et al., 2022” is not on the reference list.
· Fig 2B: What are gray circles below the protein structure scheme?
· Fig 3 title: …glandular tissues, or epithelium.
· Fig 3C: Use larger font in the insert (Fzd7 etc.).
· Fig 5 title: “in”, not “is”, right?
· Fig 5A: Larger images/fonts of p63, Brg1, Satb1 and EDC would be appreciated.
Author Response
This is an excellent review article that covers various aspects of p63, including its functions in normal and cancer cells with complexity arising from functionally distinct isoforms, its roles in senescence and aging, its mutation-associated diseases, and its role in chromatin remodeling and organization. The reference list contains recent ones and looks good as well. Minor suggestions are as follows.
Author response: We thank the reviewer for his comments. We also revised our manuscript to address the minor points suggested.
Comment 1: Figure 3 should be cited in the text where appropriate, each separately as 3A, 3B, 3C, 3D, and 3E. This comment also applies to some other figures. In terms of referring to the figures in the text, the current version is not reader-friendly.
Response 1: Thank you for pointing this out. We have referred to each panel of Figure 3 separately throughout the manuscript.
Comment 2: Related to line 157: How conserved is p63 among species, e.g., humans, mice, zebrafish etc.?
Response 2: As suggested by the reviewer, we have included details on p63 conservation among different species (lines 282-288).
Comment 3: Line 188, Tap63 to ΔNP63: Uppercase and lowercase letters to be corrected.
Response 3: We have made the necessary corrections.
Comment 4: Line 231: R298 to what amino acid?
Response 4: We appreciate the reviewer for highlighting this missing detail. The mutation change to the Q amino acid has been included in R298 (line 165).
Comment 5: Line 286: Figure 2C, not Figure 3?
Response 5: Thank you for pointing this out. We have made the necessary corrections.
Comment 6: Line 478: “Rivera-Mulia, et al., 2022” is not on the reference list.
Response 6: We thank the reviewer for pointing this out. We have added the appropriate citation to the reference list.
Comment 7: Fig 2B: What are gray circles below the protein structure scheme?
Response 7: We thank the reviewer for pointing this out. We have added appropriate labels for the gray circles in the protein structure scheme in Figure 2B.
Comment 8: Fig 3 title: …glandular tissues, or epithelium.
Response 8: Thank you for pointing this out. We have made the necessary corrections.
Comment 9: Fig 3C: Use larger font in the insert (Fzd7 etc.).
Response 9: We thank the reviewer for making this suggestion. The font size in Figure 3 has been increased to improve readability.
Comment 10: Fig 5 title: “in”, not “is”, right?
Response 10: Thank you for pointing this out. We have made the necessary corrections.
Comment 11: Fig 5A: Larger images/fonts of p63, Brg1, Satb1, and EDC would be appreciated.
Response 11: Images/fonts of p63, Brg1, Satb1, and EDC have been adjusted accordingly
Reviewer 2 Report
Comments and Suggestions for Authors
The review article titled "p63: A Master Regulator at the Crossroads Between Development, Senescence, Aging, and Cancer" by Carlos Rivera-Mulia and team provides the comprehensive literature showing the role that p63 in developmental regulation, tissue maintenance, senescence, cancer, and genome organization including their own work which shows the role of TP63 mutations in various developmental diseases. It highlights the complex role which p63 plays in various diseases, particularly in the context of development, aging, and cancer.
Overall, the review is well written and has potential to serves as a comprehensive resource for researchers interested in understanding the role and the molecular mechanisms of p63 and its potential as a therapeutic target in disease. The review has minor grammatical mistakes, i.e. in the text p63 knockout is written as p63 knockout or written as p63-/-. Authors are advised to keep symmetry. 22-hour no hyphen needed. Please increase the font size in figure 2 and 3 to make it more clear to readers.
Author Response
The review article titled "p63: A Master Regulator at the Crossroads Between Development, Senescence, Aging, and Cancer" by Carlos Rivera-Mulia and team provides the comprehensive literature showing the role that p63 in developmental regulation, tissue maintenance, senescence, cancer, and genome organization including their own work which shows the role of TP63 mutations in various developmental diseases. It highlights the complex role which p63 plays in various diseases, particularly in the context of development, aging, and cancer.
Overall, the review is well written and has potential to serves as a comprehensive resource for researchers interested in understanding the role and the molecular mechanisms of p63 and its potential as a therapeutic target in disease.
Author response: We thank the reviewer for their comments.
Comment 1: The review has minor grammatical mistakes, i.e., in the text, p63 knockout is written as p63 knockout or p63-/-. Authors are advised to keep symmetry.
Response 1: We thank the reviewer for making this suggestion. To maintain symmetry, we have replaced p63-/- with p63 knockout throughout the manuscript.
Comment 2: 22-hour no hyphen needed.
Response 2: Thank you for pointing this out. We have made the necessary corrections (line 268).
Comment 3: Please increase the font size in figure 2 and 3 to make it more clear to readers.
Response 3: We thank the reviewer for making this suggestion. The font size in Figures 2 and 3 has been increased to improve readability.
Reviewer 3 Report
Comments and Suggestions for Authors
The manuscript by Murari et al. provides a comprehensive and well-structured review of the multifaceted roles of p63 in development, tissue homeostasis, senescence, aging, and cancer. It integrates an extensive body of literature and includes a detailed discussion of TAp63 and ΔNp63 isoforms, their functional interplay, and their roles in distinct biological contexts. This discussion is thorough and well-supported by current literature. The integration of these topics is effectively organized, contributing significantly to our understanding of the overarching impact of p63 in biology. Additionally, the manuscript includes well-designed, high-quality figures that greatly enhance the reader's comprehension of complex concepts.
However, the manuscript could be further improved by addressing the following concerns:
1. While the manuscript provides comprehensive coverage of individual topics, the depth of mechanistic discussion, particularly beyond isoform-specific changes, appears limited. For example, the interaction of p63 with histone modifiers and chromatin remodeling complexes could be discussed in more detail. Expanding this discussion would provide a deeper understanding and strengthen the manuscript.
2. Given the existence of several similar reviews in recent years, the manuscript could be made more impactful and distinctive by addressing ongoing challenges in p63 research. For instance, the authors could explore hurdles in studying isoform-specific regulation and dynamic expression patterns. Including a discussion on how emerging technologies could overcome these challenges would be particularly valuable.
3. While the manuscript effectively presents a robust collection of literature, it would benefit from adding parts that synthesize this information into a cohesive narrative. This would help readers better understand the interconnected roles of p63 across different contexts and emphasize overarching themes.
4. A minor point, certain sections contain redundancy that detracts from the flow of the manuscript. For example, the roles of TAp63 and ΔNp63 in epithelial homeostasis are reiterated across multiple sections without providing new insights.
By addressing these points, the manuscript can be strengthened to provide a more distinctive and impactful review, offering deeper insights into p63’s complex biology and the challenges faced in its study.
Author Response
|
The manuscript by Murari et al. provides a comprehensive and well-structured review of the multifaceted roles of p63 in development, tissue homeostasis, senescence, aging, and cancer. It integrates an extensive body of literature and includes a detailed discussion of TAp63 and ΔNp63 isoforms, their functional interplay, and their roles in distinct biological contexts. This discussion is thorough and well-supported by current literature. The integration of these topics is effectively organized, contributing significantly to our understanding of the overarching impact of p63 in biology. Additionally, the manuscript includes well-designed, high-quality figures that greatly enhance the reader's comprehension of complex concepts. However, the manuscript could be further improved by addressing the following concerns: Author response: We thank the reviewer for their comments. We revised our manuscript to address their concerns.
Comment 1: While the manuscript provides comprehensive coverage of individual topics, the depth of mechanistic discussion, particularly beyond isoform-specific changes, appears limited. For example, the interaction of p63 with histone modifiers and chromatin remodeling complexes could be discussed in more detail. Expanding this discussion would provide a deeper understanding and strengthen the manuscript.
Response 1: We thank the reviewer for their insightful suggestion to strengthen our manuscript. We reorganized this section and expanded to discuss the potential roles of p63 in genome organization and function. Our revised manuscript also highlights the role of p63 interactions with chromatin remodelers.
|
|
Comment 2: Given the existence of several similar reviews in recent years, the manuscript could be made more impactful and distinctive by addressing ongoing challenges in p63 research. For instance, the authors could explore hurdles in studying isoform-specific regulation and dynamic expression patterns. Including a discussion on how emerging technologies could overcome these challenges would be particularly valuable. Response 2: Previous reviews on p63 indeed discuss the distinct roles of p63 on gene regulation in normal and disease stated. However, the focus of previous publications in on adult epithelial differentiation or specific roles in cancer. The motivation for writing this manuscript, was to provide an integrative review of all the complex aspects of p63 regulation. Nevertheless, we agree that a discussion of emerging technologies that can help overcoming current challenges was missing. Thus, we revised our manuscript to include the need of structural studies, as well as the opportunity that the latest advances in nucleic acid sequencing bring to enable the isoform-specific detection. |
|
|
|
Comment 3: While the manuscript effectively presents a robust collection of literature, it would benefit from adding parts that synthesize this information into a cohesive narrative. This would help readers better understand the interconnected roles of p63 across different contexts and emphasize overarching themes.
Response 3: We thank the reviewer for their comment. We have restructured the paper for a more cohesive narrative. |
|
Comment 4: A minor point, certain sections contain redundancy that detracts from the flow of the manuscript. For example, the roles of TAp63 and ΔNp63 in epithelial homeostasis are reiterated across multiple sections without providing new insights.
Response 4: Thank you for bringing this to our attention. We have removed the redundant sections from the paper, particularly sections that reiterate the roles of TAp63 and ΔNp63 in epithelial homeostasis. |
|
|
|
By addressing these points, the manuscript can be strengthened to provide a more distinctive and impactful review, offering deeper insights into p63’s complex biology and the challenges faced in its study. Author response: We thank the reviewer for their comments. We have made the requested changes.
|
Reviewer 4 Report
Comments and Suggestions for Authors
This review explores the role of p63 in developmental conditions relating to p63 mutations, epithelial cell fate decisions, chromatin remodeling, and cancer. Overall, it is a comprehensive review but there are several concerns and areas for improvement. The focus on providing a cohesive narrative, rather than diving too deeply into a single study – in this regard, more holistic view of the literature and mechanistic insights into p63 function, wherever possible, will improve the review's depth and relevance. Additionally, the review should move beyond merely describing experiments and studies. It needs to critically assess the gaps in current knowledge and propose areas for future research.
1. This review is long and meandering at places, and at times appears to lack a focus. The abstract states “This review aims to elaborate on the diverse roles of p63, focusing on disease, development, and the mechanisms controlling genome organization and function.” The review would benefit from an organized re-structuring such as discussing p63 in developmental syndromes, function in epithelial cells, then how p63 functions normally in these epithelial cells (transcriptional regulation, recruitment of coactivators, chromatin remodeling), then brief discussion how these roles can be extrapolated to cancer and potential targeting methods discussed in the literature. There are paragraphs inserted in certain places that seem out of place.
2. Discussions in some section is often heavily weighted on findings from one group. Along the same line, this review looks like a repeat of Li et al., 2023, with this review being cited numerous times to make a point.
3. The expression patterns of deltaN and TAp63 isoforms has been a bone of contention in the field for a long time. However, growing data from literature and emerging single and bulk RNA-seq datasets have clearly shown that the deltaNp63 isoforms are the primary isoforms in most epithelial cells during development and differentiation. The statement made by the authors “the TAp63 isoform is induced in early ectodermal progenitor cells expressing the markers keratins 8 (K8) and 18 (K18) and facilitates the cell fate commitment to epithelial cell lineages (Zhao et al., 2015) is incorrect. In fact, the primary findings of the paper by Zhao et al., 2015 is that TAp63 is for practical purposes undetectable during early embryonic epithelial development such as the skin epidermis. There are many publications regarding the expression of p63 isoforms that the authors can examine for more information. PMID: 34315849; PMID: 26251276; PMID: 19461998; PMID: 39404067
4. The biggest concern was the lack of insightfulness in this review; it fails to provide an adequate discussion of the literature, addressing the gaps in knowledge, and what future directions may entail. Information after information is cited without processing this information and many times, the studies cited cannot be interpreted due to improper or missing details regarding the experiments. Examples include:
a. line 231: “Of the identified TP63 mutations that cause ADULT, R298 231 results in a gain of transactivation activity of the ΔNp63γ isoform rather than a loss of DNA binding ability, although the mutation is present in the DNA binding domain (Celli 233 et al., 1999; Rinne et al., 2006).”
b. Line 502-504: “The complex cellular context in which TP63 participates in various cancers complicates parsing out exactly how TP63 dysregulation leads to cancer because the two major isoforms are expressed at different levels in different cell types and behave differently in various cancers.” Wouldn’t the distinct expression of the major p63 isoforms facilitate the dissection of their distinct roles?
c. Line 514-521: “Napoli & colleagues studied the importance of ΔNp63 for tumor initiation in LUAD and maintenance of the proliferation of distal lung stem cell populations by performing a ΔNp63-knockout in LUAD mice. ΔNp63-knockout mice exhibited a twofold decrease in hyperplastic lung lesions, and immunofluorescence confirmed that tumors from these mice contained fewer cells positive for the distal lung stem cell and LUAD markers CCSP and SPC compared to the controls, suggesting that ΔNp63 is essential for tumorigenesis in LUAD and acts to maintain the proliferation of distal lung stem cell populations.” Lines 574-577 pertaining to SCC: “In Ramsey et al. 2013, the authors first induce tumors in the mice, allowing these tumors to progress, and then perform the conditional KO of TP63 and observe tumor regression after TP63 KO (Ramsey et al., 576 2013). In Lakshmanachetty et al. 2019, the authors first perform conditional KO of TP63 and then induce tumors in the mice (Lakshmanachetty et al., 2019).” crucial aspect that deserves mention is the oscillatory expression of DeltaNp63 during cancer progression PMID: 32312834. This is a good opportunity to tie the evidence together to suggest a conclusion; so far this review does a poor job at connecting the dots.
d. Line 564-565: “…expression of p63 is not required for oncogenic outcomes in SCC subtypes and that multiple pathways may contribute to cancer progression (Lakshmanachetty et al. 2019).” The molecular subtypes are not discussed here and how p63 is involved therein.
e. Line 639-641: “Similar to other cancers, the current literature shows that TP63 may play contradictory roles in breast cancer, which may be due to the differential expression of the two isoforms of TP63 in different tumors.” This is also a missed opportunity where the authors failed to bring back luminal vs basal breast cancers (which the authors discuss in lines 317-320 to exhibit distinct p63 isoform expression) enriched for different isoforms and how tumors arising from either or may reflect the contradictory role of p63 in breast cancer.
f. The sections of this review discussing the role of p63 in chromatin remodeling is not organized and fails to highlight the significant role of p63 as a pioneer factor though a model is provided for this role in figure 5.
g. Line 875-877: “TAp63 controls mostly early progenitors, and dysregulation can alter these progenitors and their derivative populations. ∆Np63 is expressed primarily in the fully differentiated cells and controls proliferation.” Basal cells of the epithelium are not fully differentiated, did the authors mean committed cells?
5. The figures in this review look great but some figures do not provide much information and are often based solely on one study, are mislabeled, or could be expanded.
a. The aphlafold3 structure of p63 bound to consensus DNA sequences is shown in figure 1 but the description of this figure or main text does not describe what is being shown (is this a dimer of the Tap63 alpha isoform?). The second panel in figure 1B is said to show “the tetrameric complex of p63 … In agreement with the experimental evidence, the model predicts interactions between the C-terminal transactivation 102 domain (TA1) and the inhibitory domain (ID). However, this panel does not show this interaction perhaps due to the view of the structure. Also, this appears to be a dimer complex, and not a tetrameric complex. Additionally, the N-terminal domain of the full-length Tap63 isoform is the TA1 domain, not the C-terminal portion of the protein as labeled in the figure and mentioned in the main text. This figure could benefit from showing the tetrameric complexes that can form between p63 isoform and p53/p73. Finally, the authors don’t describe what they mean by full-length p63 protein sequence and what is the consensus binding motif that was used.
b. Line 128: “Once the epithelial basal layer is formed, ΔNp63 expression is induced to trigger the stratification of the epithelium, generating additional layers that express keratins, and its expression is maintained in the basal layer of stratified epithelial tissue (Figure 2)”. ΔNp63 expression is enriched within the basal epithelial but it is not only maintained within the basal layer; its expression exhibits a gradient in the stratified epithelium. When a figure is referenced after a statement, the figure should match what the statement says.
c. Figure 4 describes the balance between the p63 main isoforms in regulating proliferation in keratinocytes and senescence in fibroblasts from two respective studies, though the focus of the review sems to be on epithelial cells. Based on existing RNA-seq data, it is clear that endogenous expression of p63 in fibroblasts is minimal.
6. The conclusion summarizes everything discussed in the review but provides no discussion on the implication of these findings.
7. Random statements and points are raised throughout the manuscript that do not flow with the main points in that section. Examples include “Aberrant p63 overexpression in the endometrium has also been observed in the case of endometriosis (Pashaei et al., 2022).” “p63 deficiency also impairs cardiac differentiation (Rouleau et al., 2011).”
8. Some important information relevant to this review were not mentioned.
a. Line 89 – 91: “For instance, while the TAp63𝛼 isoform represses cell proliferation, the ΔNp63𝛼 isoform promotes cell proliferation and survival (Chen et al. 2018). However, the molecular complexity of TP63 makes it challenging to dissect the specific mechanisms involved in each pathway.” What about the intramolecular interaction of p63 isoforms and between p63 and p53, p73?
Author Response
|
This review explores the role of p63 in developmental conditions relating to p63 mutations, epithelial cell fate decisions, chromatin remodeling, and cancer. Overall, it is a comprehensive review but there are several concerns and areas for improvement. The focus on providing a cohesive narrative, rather than diving too deeply into a single study – in this regard, more holistic view of the literature and mechanistic insights into p63 function, wherever possible, will improve the review's depth and relevance. Additionally, the review should move beyond merely describing experiments and studies. It needs to critically assess the gaps in current knowledge and propose areas for future research.
Author response: We thank the reviewer for their insightful comments. We agree with his assessment and have thoroughly revised our manuscript to address their concerns.
Comment 1: This review is long and meandering at places, and at times appears to lack a focus. The abstract states “This review aims to elaborate on the diverse roles of p63, focusing on disease, development, and the mechanisms controlling genome organization and function.” The review would benefit from an organized re-structuring such as discussing p63 in developmental syndromes, function in epithelial cells, then how p63 functions normally in these epithelial cells (transcriptional regulation, recruitment of coactivators, chromatin remodeling), then brief discussion how these roles can be extrapolated to cancer and potential targeting methods discussed in the literature. There are paragraphs inserted in certain places that seem out of place.
Response 1: We thank the reviewer for their comment. This manuscript started as a scientific writing exercise for undergraduate and high-school students (listed as authors), it was later edited by the graduate students and the corresponding author. Nevertheless, we agree with the reviewer in that major changes were needed and have restructured our manuscript for a more cohesive narrative, removed redundance, and focused on the concepts being discussed. |
|
|
|
Comment 2: Discussions in some section is often heavily weighted on findings from one group. Along the same line, this review looks like a repeat of Li et al., 2023, with this review being cited numerous times to make a point.
Response 2: We thank the reviewer for bringing this to our attention. We have revised our manuscript thoroughly and used more appropriate references.
|
|
Comment 3: The expression patterns of deltaN and TAp63 isoforms has been a bone of contention in the field for a long time. However, growing data from literature and emerging single and bulk RNA-seq datasets have clearly shown that the deltaNp63 isoforms are the primary isoforms in most epithelial cells during development and differentiation. The statement made by the authors “the TAp63 isoform is induced in early ectodermal progenitor cells expressing the markers keratins 8 (K8) and 18 (K18) and facilitates the cell fate commitment to epithelial cell lineages (Zhao et al., 2015) is incorrect. In fact, the primary findings of the paper by Zhao et al., 2015 is that TAp63 is for practical purposes undetectable during early embryonic epithelial development such as the skin epidermis. There are many publications regarding the expression of p63 isoforms that the authors can examine for more information. PMID: 34315849; PMID: 26251276; PMID: 19461998; PMID: 39404067
Response 3: We thank the reviewer for their comment. We have revised our manuscript to discuss the expression patterns of p63 isoforms with accuracy. We have also added the suggested papers throughout the review when appropriate. PMID: 34315849 - lines 244-246; PMID: 26251276 - lines 209-212; PMID:19461998 - lines 235-238; PMID: 39404067. |
|
|
|
Comment 4: The biggest concern was the lack of insightfulness in this review; it fails to provide an adequate discussion of the literature, addressing the gaps in knowledge, and what future directions may entail. Information after information is cited without processing this information and many times, the studies cited cannot be interpreted due to improper or missing details regarding the experiments. Examples include:
Response 4: We thank the reviewer for their assessment and suggestions to improve our manuscript. Below are our responses for each of the specific points.
|
|
Comment 4a: Line 231: “Of the identified TP63 mutations that cause ADULT, R298 results in a gain of transactivation activity of the ΔNp63γ isoform rather than a loss of DNA binding ability, although the mutation is present in the DNA binding domain (Celli 233 et al., 1999; Rinne et al., 2006).”
Response 4a: We appreciate the reviewer for highlighting this issue. We added a discussion of the annotated mutations, potential effects, and the need for further characterization of these mutations.
Comment 4b: Line 502-504: “The complex cellular context in which TP63 participates in various cancers complicates parsing out exactly how TP63 dysregulation leads to cancer because the two major isoforms are expressed at different levels in different cell types and behave differently in various cancers.” Wouldn’t the distinct expression of the major p63 isoforms facilitate the dissection of their distinct roles?
Response 4b: We thank the reviewer for their comment. We have rephrased these lines to imply that the distinct expression of the major isoform p63 might help facilitate the dissection of their distinct roles.
Comment 4c: Line 514-521: “Napoli & colleagues studied the importance of ΔNp63 for tumor initiation in LUAD and maintenance of the proliferation of distal lung stem cell populations by performing a ΔNp63-knockout in LUAD mice. ΔNp63-knockout mice exhibited a twofold decrease in hyperplastic lung lesions, and immunofluorescence confirmed that tumors from these mice contained fewer cells positive for the distal lung stem cell and LUAD markers CCSP and SPC compared to the controls, suggesting that ΔNp63 is essential for tumorigenesis in LUAD and acts to maintain the proliferation of distal lung stem cell populations.” Lines 574-577 pertaining to SCC: “In Ramsey et al. 2013, the authors first induce tumors in the mice, allowing these tumors to progress, and then perform the conditional KO of TP63 and observe tumor regression after TP63 KO (Ramsey et al., 576 2013). In Lakshmanachetty et al. 2019, the authors first perform conditional KO of TP63 and then induce tumors in the mice (Lakshmanachetty et al., 2019).” crucial aspect that deserves mention is the oscillatory expression of DeltaNp63 during cancer progression PMID: 32312834. This is a good opportunity to tie the evidence together to suggest a conclusion; so far this review does a poor job at connecting the dots.
Response 4c: We thank the reviewer for pointing out this missed opportunity. We referred to the suggested paper and condensed this section, made sure we have a cohesive narrative, and connected these observations.
Comment 4d: Line 564-565: “…expression of p63 is not required for oncogenic outcomes in SCC subtypes and that multiple pathways may contribute to cancer progression (Lakshmanachetty et al. 2019).” The molecular subtypes are not discussed here and how p63 is involved therein. Response 4d: We have revised our manuscript to specify the site-specific molecular subtypes of SCC.
Comment 4e: Line 639-641: “Similar to other cancers, the current literature shows that TP63 may play contradictory roles in breast cancer, which may be due to the differential expression of the two isoforms of TP63 in different tumors.” This is also a missed opportunity where the authors failed to bring back luminal vs basal breast cancers (which the authors discuss in lines 317-320 to exhibit distinct p63 isoform expression) enriched for different isoforms and how tumors arising from either or may reflect the contradictory role of p63 in breast cancer.
Response 4e: We have rewritten certain parts of the breast cancer section to address this concern.
Comment 4g: The sections of this review discussing the role of p63 in chromatin remodeling is not organized and fails to highlight the significant role of p63 as a pioneer factor though a model is provided for this role in figure 5.
Response 4g: We have revised our manuscript to better describe the findings on p63 control of chromatin remodeling, highlighted its role as pioneer factor, and updated the figure accordingly.
Comment 5: The figures in this review look great but some figures do not provide much information and are often based solely on one study, are mislabeled, or could be expanded.
Response 5: We thank the reviewer for these comments. We have revised our manuscript and edited the figures and their legends to better reflect the concepts.
Comment 5a: The aphlafold3 structure of p63 bound to consensus DNA sequences is shown in figure 1 but the description of this figure or main text does not describe what is being shown (is this a dimer of the Tap63 alpha isoform?). The second panel in figure 1B is said to show “the tetrameric complex of p63 … In agreement with the experimental evidence, the model predicts interactions between the C-terminal transactivation 102 domain (TA1) and the inhibitory domain (ID). However, this panel does not show this interaction perhaps due to the view of the structure. Also, this appears to be a dimer complex, and not a tetrameric complex. Additionally, the N-terminal domain of the full-length Tap63 isoform is the TA1 domain, not the C-terminal portion of the protein as labeled in the figure and mentioned in the main text. This figure could benefit from showing the tetrameric complexes that can form between p63 isoform and p53/p73. Finally, the authors don’t describe what they mean by full-length p63 protein sequence and what is the consensus binding motif that was used.
Response 5a: We have added all the detailed information on the DNA sequence of the consensus binding motif and the use of four copies of the TAp63 isoform as input for the prediction. We also thank the reviewer for catching the mislabeling of the C-terminal and N-terminal regions of the proteins. In addition, we are providing the PyMol file as supplementary material to facilitate its use by the community.
Comment 5b: Line 128: “Once the epithelial basal layer is formed, ΔNp63 expression is induced to trigger the stratification of the epithelium, generating additional layers that express keratins, and its expression is maintained in the basal layer of stratified epithelial tissue (Figure 2)”. ΔNp63 expression is enriched within the basal epithelial but it is not only maintained within the basal layer; its expression exhibits a gradient in the stratified epithelium. When a figure is referenced after a statement, the figure should match what the statement says.
Response 5b: We edited figure 2 to include the reported expression gradient of ∆Np63.
Comment 5c: Figure 4 describes the balance between the p63 main isoforms in regulating proliferation in keratinocytes and senescence in fibroblasts from two respective studies, though the focus of the review seems to be on epithelial cells. Based on existing RNA-seq data, it is clear that endogenous expression of p63 in fibroblasts is minimal.
Response 5c: Correct, expression patterns of p63 isoforms have been characterized mostly on keratinocytes while effects on senescence response come from distinct cell types and is unclear what are the specific roles in distinct contexts. We removed this figure from our manuscript.
|
|
Comment 6: The conclusion summarizes everything discussed in the review but provides no discussion on the implication of these findings.
Response 6: We expanded our conclusion section to discuss implications and future directions.
|
|
Comment 7: Random statements and points are raised throughout the manuscript that do not flow with the main points in that section. Examples include “Aberrant p63 overexpression in the endometrium has also been observed in the case of endometriosis (Pashaei et al., 2022).” “p63 deficiency also impairs cardiac differentiation (Rouleau et al., 2011).”
Response 7: We thank the reviewer for their insights. We have removed random statements that distract from the flow of the main points of the paper. |
|
Comment 8: Some important information relevant to this review were not mentioned. Comment 8a: Line 89 – 91: “For instance, while the TAp63𝛼 isoform represses cell proliferation, the ΔNp63𝛼 isoform promotes cell proliferation and survival (Chen et al. 2018). However, the molecular complexity of TP63 makes it challenging to dissect the specific mechanisms involved in each pathway.” What about the intramolecular interaction of p63 isoforms and between p63 and p53, p73?
Response 8a: We thank the reviewers for their insights. We have included details on the intramolecular interactions of p53/p63/p73. |
Round 2
Reviewer 4 Report
Comments and Suggestions for Authors
The revised manuscript though improved still suffers from major problems. It seems that the authors are not doing a diligent job of vetting their manuscript and making sure that the Review that they have put together reflects what is known in the field. The burden should not be on the reviewer to correct the manuscript. Here are 3 examples (there are many many others need attention)
Example 1: The developmental control mechanisms of p63 still need to be fully understood, but evidence is being unveiled in mouse models. Furthermore, p63 is required for both male and female germline maintenance (Gonfloni et al., 2009; Kerr et al., 2012; Lena et al., 2021; Suh et al., 2006). In adult tissues p63 is also pivotal for maintaining epidermal homeostasis (Ma & Tian et al., 2014). As a transcription factor, it regulates cellular pathways required for adult tissue survival, self renewal, and differentiation (Li et al., 2023).
Review: The above portion taken from the Introduction section seems to have random sentences and ideas that are thrown together in an incoherent fashion. All the references shown are for p63’s role in female germline and not for male. The Ma and Tian reference paper is not at all ideal or suitable for addressing the role of p63 and epidermal homeostasis.
Example 2: The first ectodermal dysplasia case was reported in 1848 (Thurnam, 1848; Weech, 1929). Further studies into the molecular basis implicated multiple missense mutations of p63. R304W is the most well-characterized (Biwei et al., 2022).
Review: The review is about p63, so not sure whether a reference to an 1848 paper on the first case of ectodermal dysplasia fits well. The sentence “R304W is the most well-characterized” does not really convey anything meaningful and the Biwei et al., reference is just a case study.
Example 3: The maintenance of other epithelial appendages, such as hair, is also regulated by p63 activity (Nelson, 2016). Stem cell populations located within several niches of the hair follicle act to maintain the epithelium through the activity of ΔNp63𝛼, which is expressed only in these stem cells as well as the outer root sheath and matrix cells (Truong et al., 2006). The hair follicle stem cell niche was depleted in mice with mutant ΔNp63 (Romano 381 2010). Additionally, mice with p63 knock-out produce progeny without hair follicles, illustrating the critical role p63 plays in developing epithelial appendages (Nelson, 2016).
Review: The above paragraph is an illustration of how a specific topic is dealt with superficially in this review. In this case some of the references provided for p63 and hair follicles are completely irrelevant and wrong. For e.g., Truong et al paper is all about human keratinocytes and has nothing to do with hair follicles. Same case with Nelson et al., which deals superficially with wound-induced hair neogenesis.
Author Response
Comment 1: The revised manuscript though improved still suffers from major problems. It seems that the authors are not doing a diligent job of vetting their manuscript and making sure that the Review that they have put together reflects what is known in the field. The burden should not be on the reviewer to correct the manuscript. Here are 3 examples (there are many many others need attention).
Response: We have revised our manuscript thoroughly and ensured the appropriate references highlight the latest advances in the field. Below, we explain how we addressed the specific examples raised by the reviewer. In addition, multiple other changes have been highlighted in the main text of our revised manuscript to address this concern.
Example 1: The developmental control mechanisms of p63 still need to be fully understood, but evidence is being unveiled in mouse models. Furthermore, p63 is required for both male and female germline maintenance (Gonfloni et al., 2009; Kerr et al., 2012; Lena et al., 2021; Suh et al., 2006). In adult tissues p63 is also pivotal for maintaining epidermal homeostasis (Ma & Tian et al., 2014). As a transcription factor, it regulates cellular pathways required for adult tissue survival, self renewal, and differentiation (Li et al., 2023).
Review: The above portion taken from the Introduction section seems to have random sentences and ideas that are thrown together in an incoherent fashion. All the references shown are for p63’s role in female germline and not for male. The Ma and Tian reference paper is not at all ideal or suitable for addressing the role of p63 and epidermal homeostasis.
Response: We revised our manuscript and edited this part accordingly. Appropriate references on the role of p63 in male germline have been added, as well as more appropriate references for epidermal homeostasis.
Example 2: The first ectodermal dysplasia case was reported in 1848 (Thurnam, 1848; Weech, 1929). Further studies into the molecular basis implicated multiple missense mutations of p63. R304W is the most well-characterized (Biwei et al., 2022).
Review: The review is about p63, so not sure whether a reference to an 1848 paper on the first case of ectodermal dysplasia fits well. The sentence “R304W is the most well-characterized” does not really convey anything meaningful and the Biwei et al., reference is just a case study.
Response: the motivation for including the publication from 1848 was to show the readers for how long these syndromes have been reported. However, reviewer is correct in that this reference does not inform on the molecular mechanisms associated with p63. Thus, we have remove this reference as well as other case studies. In addition, we have clarified that the R304W mutation plays a critical role in the disease and that it has been targeted to rescue p63 function in cells from these patients.
Example 3: The maintenance of other epithelial appendages, such as hair, is also regulated by p63 activity (Nelson, 2016). Stem cell populations located within several niches of the hair follicle act to maintain the epithelium through the activity of ΔNp63?, which is expressed only in these stem cells as well as the outer root sheath and matrix cells (Truong et al., 2006). The hair follicle stem cell niche was depleted in mice with mutant ΔNp63 (Romano 381 2010). Additionally, mice with p63 knock-out produce progeny without hair follicles, illustrating the critical role p63 plays in developing epithelial appendages (Nelson, 2016).
Review: The above paragraph is an illustration of how a specific topic is dealt with superficially in this review. In this case some of the references provided for p63 and hair follicles are completely irrelevant and wrong. For e.g., Truong et al paper is all about human keratinocytes and has nothing to do with hair follicles. Same case with Nelson et al., which deals superficially with wound-induced hair neogenesis.
Response: We revised our manuscript and replaced the references with the specific studies characterizing the role of p63 in the hair follicles.